# Artificially cloaked viral nanovaccine for cancer immunotherapy

Manlio Fusciello[1,8], Flavia Fontana [2,8], Siri Tähtinen[1], Cristian Capasso[1], Sara Feola [1], Beatriz Martins[1], Jacopo Chiaro[1], Karita Peltonen [1], Leena Ylösmäki[1], Erkko Ylösmäki[1], Firas Hamdan [1], Otto K. Kari[1], Joseph Ndika [3], Harri Alenius[3,4], Arto Urtti[1,5], Jouni T. Hirvonen[2], Hélder A. Santos [2,6]* & Vincenzo Cerullo [1,6,7]*

Virus-based cancer vaccines are nowadays considered an interesting approach in the field of cancer immunotherapy, despite the observation that the majority of the immune responses they elicit are against the virus and not against the tumor. In contrast, targeting tumor associated antigens is effective, however the identification of these antigens remains challenging. Here, we describe ExtraCRAd, a multi-vaccination strategy focused on an oncolytic virus artificially wrapped with tumor cancer membranes carrying tumor antigens. We demonstrate that ExtraCRAd displays increased infectivity and oncolytic effect in vitro and in vivo. We show that this nanoparticle platform controls the growth of aggressive melanoma and lung tumors in vivo both in preventive and therapeutic setting, creating a highly specific anti-cancer immune response. In conclusion, ExtraCRAd might serve as the next generation of personalized cancer vaccines with enhanced features over standard vaccination regimens, representing an alternative way to target cancer.

[1] Drug Research Program, Division of Pharmaceutical Biosciences and Digital Precision Cancer Medicine Flagship (iCAN), Faculty of Pharmacy, University of Helsinki, FI-00140 Helsinki, Finland. [2] Drug Research Program, Division of Pharmaceutical Chemistry and Technology, Faculty of Pharmacy, University of Helsinki, FI-00140 Helsinki, Finland. [3] Systems Toxicology Group, Department of Bacteriology and Immunology, Medicum, University of Helsinki, FI-00140 Helsinki, Finland. [4] Institute of Environmental Medicine, Karolinska Institutet, SE-171 77 Stockholm, Sweden. [5] School of Pharmacy, Faculty of Health Sciences, University of Eastern Finland, FI-70211 Kuopio, Finland. [6] Helsinki Institute of Life Science (HiLIFE), University of Helsinki, FI-00014 Helsinki, Finland. [7] Department of Molecular Medicine and Medical Biotechnology and CEINGE, Naples University Federico II, 80131 Naples, Italy. [8] These authors contributed equally: Manlio Fusciello, Flavia Fontana. *email: helder.santos@helsinki.fi; vincenzo.cerullo@helsinki.fi

In the last decade, research on cancer immunotherapy resulted in a new set of potential treatments with promising results in the clinics[1]. Among these, immune check-point inhibitors are one of the few immunotherapies that have been clinically validated, yet with variable results, ranging from complete responses to hyperprogression[2–4].

Among the different experimental treatments, active cancer immunotherapy and, more specifically, viral therapy hold great promises for the future[5,6]. Theoretically, tumor lysate can elicit anti-tumor activity by presenting the immune system with a wide range of tumor-associated antigens (TAAs) and neoantigens for immune system processing[7]. Unfortunately, when tumor lysate lacks a proper stimulation, it could drive to tolerance by the immune system, resulting in an ineffective approach[8].

Conversely, viruses are recognized as non-self, thereby leading to the initiation of an immune response due to their natural adjuvant properties[9]. These attributes can be exploited in cancer immunotherapy by the use of oncolytic viruses, which are viruses that, by design, only replicate in cancer cells, while leaving the healthy cells unharmed. The viral replication results in tumor cell lysis and in the release of tumor antigens in the tumor microenvironment[10]. These tumor antigens can be then taken up by antigen presenting cells (APC) and utilized to direct the immune response against the rightful target, the cancer cells[11].

Ideally, after intratumoral administration of an oncolytic virus, sufficient amounts of tumor antigens are released and picked up by dendritic cells (DCs) to present them to T cells in lymphoid organs, eventually leading up to an anti-tumor immune response[9]. However, administration of naked oncolytic virus often favors the induction of anti-viral over anti-tumor immunity. To overcome this hurdle, vaccination strategies with administration of viruses coated with peptides derived from TAAs have been found to re-direct the immune response against the tumor and enhance the therapeutic efficacy oncolytic viruses[12,13].

However, finding the right peptides to attach on the virus surface is a complex process that involves screening of the tumor peptides, fishing out from wide pools of candidates to create the perfect recipe to precisely and promptly activate pools of T cells against the target cells. Moreover, peptides selection is not always possible and the current state-of-the-art technology does not allow scientists and clinicians to have clear indications on which epitopes to use and from which proteins. Thereby, new and efficient approaches are needed to develop the best tools in order to exploit the efficacy and precision of immunotherapies.

Here, we seek to develop a technology to equip an oncolytic adenovirus with the pool of antigens, mirroring the tumor mass, leading to the creation of a viral nanoparticle (Extra conditionally replicating adenoviruses, ExtraCRAd). The first step involves the extraction of cell membrane from well-characterized cancer cell lines, followed by coating the virus with the cell membrane through a membrane extrusion process[14]. This technology shows significant results in slowing tumor growth down, activating specific anti-tumoral response in both therapeutic and vaccination set-up in different murine tumor models.

## Results

**Generation and characterization of ExtraCRAd.** Firstly, we sought out to confirm the successful encapsulation of the oncolytic virus within the cancer cell membrane, as hypothesize in Fig. 1. To this end, the biohybrid viral nanoparticle ExtraCRAd was imaged by cryo-transmission electron microscopy (TEM). As shown in Fig. 2a, the co-extrusion of virus and cancer cell membrane resulted in an artificially enveloped virus (Fig. 2a, c). The extrusion process produced both individually enveloped and

groups of viruses enveloped within one membrane vesicle. Nanotracking analysis (NTA) served to investigate the formation of new populations of particles when the membranes were combined with the virus (Fig. 2b, c). This confirmed the qualitative results of the cryo-TEM. As shown in Fig. 2b, bare viruses are characterized by a size of $107.0 \pm 6.1$ nm, defined by one single peak, signal of a homogenous particles population. As for the cell membrane vesicles, they resulted slightly smaller compared to the virus particles, with a diameter of $92.8 \pm 0.5$ nm, retaining high homogeneity. Finally, ExtraCRAd showed a consistent population with size of $117.7 \pm 0.8$ nm, representing a single viral particle enveloped within the 10 nm cell membrane layer[15]. However, NTA analysis (presented in detail in Supplementary Fig. 1) also revealed the presence of several peaks corresponding to the formation of populations characterized by bigger size; we hypothesize that these populations correspond to aggregates of viral particles enveloped by a single membrane (as evident also from the cryo-TEM pictures) formed due to the pore size of the membrane selected for the extrusion (800 nm). This pore size was initially chosen to avoid any harm to the viral particles during the process.

Furthermore, the NTA analysis enabled us to quantify the amount of particles in the different samples (adenovirus particles, cell membrane vesicles, and ExtraCRAd); the number of particles produced by the lysis of $3 \times 10^6$ cells, followed by ultracentrifugation and extrusion is 40-fold (vesicles) and tenfold (ExtraCRAd) higher than the number of viral particles (Fig. 2c). The significant decrease in the number of particles between the sample of cell membranes alone and ExtraCRAd is indicative of the membrane assembling process around the viral particles occurring during the extrusion process, with the creation of a stable final product. The difference between the number of events recorded in the virus sample and in ExtraCRAd is however not statistically significant.

To investigate the effect of different physiological buffers on the extrusion process, dynamic light scattering (DLS) and electrophoretic light scattering (ELS) were employed to check size of the virus (Fig. 2d), homogeneity and surface charge (polydispersity index (PdI) and zeta (ζ)-potential values for the samples in physiological buffers are presented in Supplementary Fig. 2b, c). The average hydrodynamic diameter for ExtraCRAd extruded in isotonic glucose (5.4%) isotonic solution or in saline 0.9% solution ranges, respectively, from $526 \pm 39$ nm for saline to microsized aggregates ($1504 \pm 85$ nm) for glucose, highlighting the unsuitability of these buffers in the extrusion process (Supplementary Fig. 2a). As for phosphate-buffered saline (PBS), the initial size of the virus not coated ($115.8 \pm 0.8$ nm) increases after extrusion with the cell membrane to $122.9 \pm 0.8$ nm. We also extruded the samples in ultrapure (milli-Q) water, where the virus before extrusion presented an average hydrodynamic diameter of $133.6 \pm 2.9$ nm, which increased after extrusion to $140.9 \pm 2.7$ nm. As for the homogeneity of the samples, as reported in Supplementary Fig. 2b, the extrusion in glucose (5.4%) resulted in a widely polydisperse sample (PdI $0.439 \pm 0.03$), while for the other extrusion buffer assessed the sample produced retained a high homogeneity, as indicated by the values of PdI below 0.2 (0.206 for ultrapure water, 0.140 for PBS, and 0.195 for saline solution). Moreover, the initial population of virus, as highlighted also in the NTA analysis, was highly monodisperse, with PdI of 0.152 when dispersed in ultrapure water and 0.084 when suspended in PBS. The surface charge of ExtraCRAd was highly negative (ranging from $-13$ to $-22$ mV) for the samples extruded in ultrapure water, PBS, and saline solution, while it was less negative ($-7.9$ mV) for samples extruded in glucose (5.4%; Supplementary Fig. 2c).

Finally, in order to evaluate the feasibility of the cell membrane as antigenic source, we analyzed ExtraCRAd (using virus and

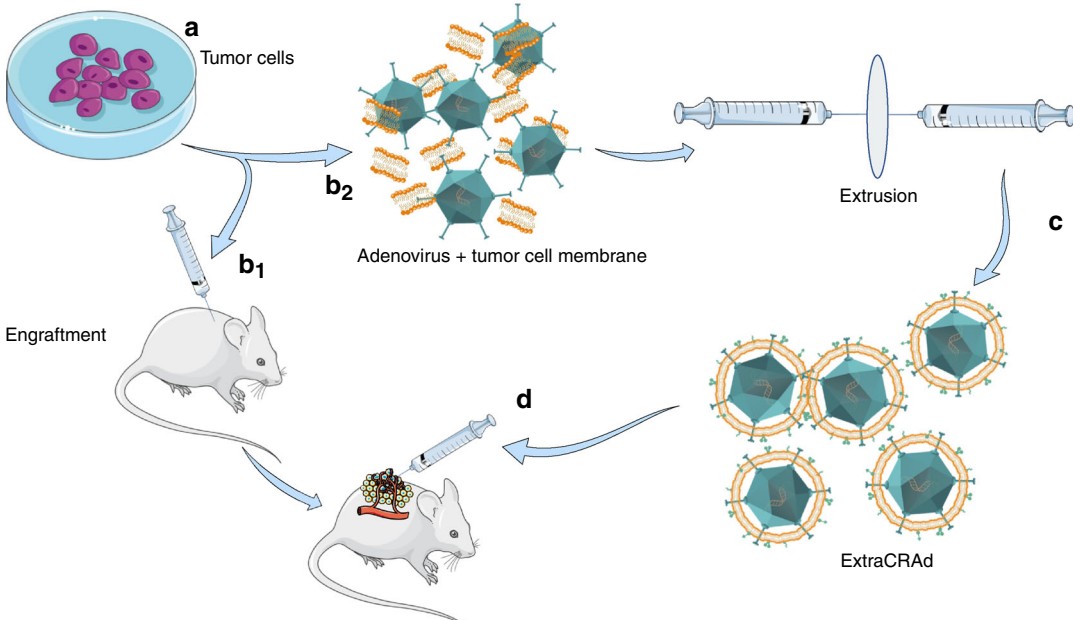

**Fig. 1 ExtraCRAd, production and treatment.** Tumor cells (**a**) were cultured and engrafted into mouse model (**b₁**). We employed the same cell line to extract the cell membrane: cells were lysed, tumor membrane isolated and mixed with an oncolytic adenovirus serotype 5, with a 24-base-pairs deletion, carrying -CpG islands (i.e., A5-Δ24-CpG)[60] (**b₂**). Through the extrusion process the virus was artificially wrapped with the cancer cell membrane to obtain ExtraCRAd (**c**). Finally, established tumors were treated with multiple intratumoral injections of ExtraCRAd (**d**). The petri dish, mice and syringes images are taken from Servier Medical Art, under Creative Commons Attribution 3.0 Unported License.

extruded cell membrane vesicles as controls) by proteomics approach. As displayed in Fig. 2e, a sample of virus presents only seven human proteins associated with ribosomes, and possibly derived from the production of the virus in human cell lines or from other contaminations, derived from the production equipment. Most importantly, both extruded membrane vesicles and ExtraCRAd displayed a similar number of total hits, with 125 proteins (corresponding to 94.7% of the total protein hits in membranes and 88.1% for ExtraCRAd) in common between the two samples. Moreover, as presented in Supplementary Fig. 3, based on subsequent bioinformatics of the mass spectrometry results, some of the proteins recovered on the cell membrane are involved in the presentation of antigens (major histocompatibility, MHC, complexes) (Supplementary Fig. 4). The most abundant protein recovered was the major vault protein (MVP), implicated in various drug resistance mechanisms in cancer cells[16]. These results confirmed that the virus was effectively encapsulated within a layer of cell membrane. Furthermore, the presence of protein hits on the cell membrane wrapped around the virus remained similar as to a control extruded cell membrane. We hypothesize that these proteins could be considered tumor-antigens and they could eventually direct the anti-tumor immune response.

Based on the physical characterization described above, we hypothesized the structure of ExtraCRAd as shown in Fig. 2f, where a core constituted of one (or several) viral particles is enveloped by a lipid bilayer derived from the cell membrane of cancer cells. This membrane brings along membrane proteins that will acts as antigens, priming the immune system against the tumor.

**ExtraCRAd displays increased infectivity in relevant cancer cells.** After successfully proving the encapsulation of the virus within the membrane, we investigated whether the extrusion process resulted in any damage to viral infectivity. The infectious titer of adenovirus decreased from $2.24 \times 10^{11}$ pfu/ml for naive

virus to $1.92 \times 10^{11}$ pfu/ml after extrusion, equivalent to a 15% decrease in infectivity. We then evaluated the functionality of the virus after the coating process with the cell membrane. We compared the naked oncolytic adenovirus to ExtraCRAd in cell viability assay using two human cancer cell lines, A549 and SKOV-3-Luc, characterized by different levels of expression of the human Coxsackie and Adenovirus receptor (CAR). A549 exhibits a high level of CAR receptors; after 2 h of infection, followed by 3 days of incubation, we obtained a statistically significant difference between cells infected with naked adenovirus or ExtraCRAd (Fig. 3a). The infection with ExtraCRAd at 100 and 10 multiplicity of infection (MOI) results in a 90% reduction in the viability (compared to a 60% reduction at 100 MOI and 10% reduction at 10 MOI for the naked virus). The increased cytopathic effect of ExtraCRAd is seen also at lower MOI (1), with a 50% decrease in the cell viability. In contrast, SKOV-3 cell line is characterized by a low amount of CAR receptors that hinders the entry of the virus within the cell. This is reflected by the lower killing activity of the naked adenovirus, as shown in Fig. 3b (blue line). However, when ExtraCRAd is administrated to the cells, the 3-day viability of SKOV-3 cells is reduced by 25% compared to the naked virus. These results suggest that there are differences in the uptake mechanism between naked and coated virus (i.e., the property of the cancer membrane to fuse with the membrane of the cells and deliver the virus intracellularly overcoming the need for a CAR receptor and the uptake of ExtraCRAd as a nanoparticle through endocytosis)[17]. As shown in Supplementary Fig. 5 ExtraCRAd increased the viral infectivity significantly in both low- and high-CAR cell lines bypassing the standard mechanism of infection through the viral receptor.

We then evaluated the oncolysis in a more complex model, by implanting human xenografts in nude mice. The results presented in Fig. 3c show statistically significant increase in the oncolysis of established human xenograft tumors after a single treatment at day 15 with ExtraCRAd when compared to the naked adenovirus. This difference is accentuated with the second administration of

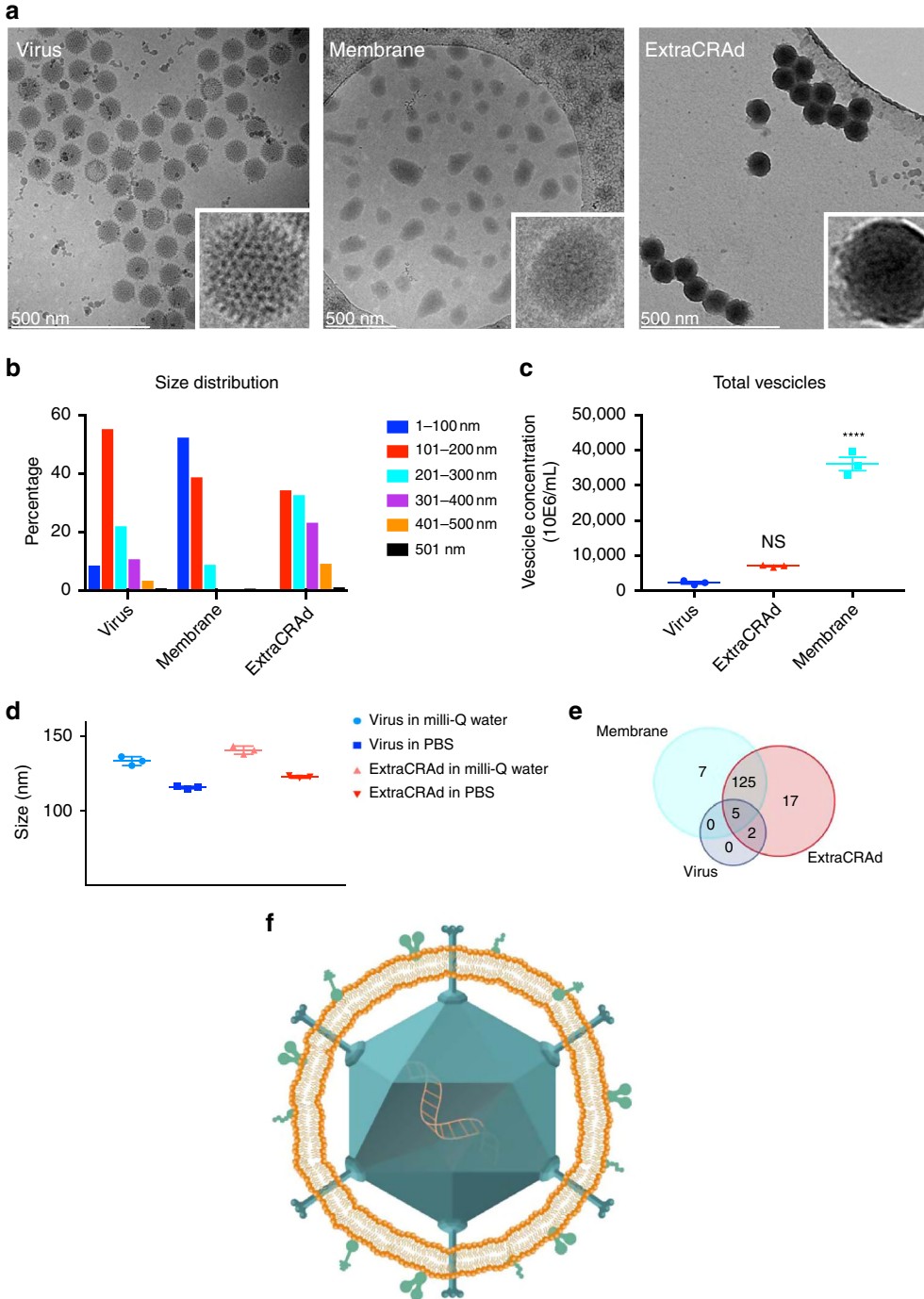

**Fig. 2 Physical characterization of ExtraCRAd.** A Cryo-transmission electron microscopy (TEM) images of **a** virus, **b** lipid cancer membrane vesicles, and **c** ExtraCRAd. B16.OVA cell line was employed as source for the membrane. Scale bar: 500 nm. The insets on the right were magnified ten times compared to the original image. **b** Nano-tracking analysis (NTA) showing the size distribution of the samples. The size distribution is expressed as percentage of the total population. The size intervals are presented on the right. **c** NTA quantification of the concentration of viral particles or nanosized vesicles in the samples. The results are presented as number × $10^6$ vesicles per ml. The data were analyzed by two-way ANOVA followed by Tukey's post-test. The level of statistical significance was set at probability level ****$p < 0.0001$. **d** Dynamic light scattering (DLS) size analysis of the system in different extrusion buffers. The results are presented as mean ± s.d. ($n = 3$). **e** Venn-diagram representing the results retrieved from the mass spectrometry analysis of the samples (virus, ExtraCRAd produced with membranes derived from human lung cancer A549 cells and extruded membrane vesicles from A549 cells). The Venn size reflects the total number of unique proteoforms for each sample, with the shared proteins presented in the intersection of the diagrams. **f** Graphic illustration of an ExtraCRAd: adenovirus 5-D24-CpG (light blue), lipid membrane (orange), and proteins (light green). The error bars indicate s.d.

the treatment at day 25. More importantly, these results are in agreement with the in vitro killing assays, highlighting the promising oncolytic effect of the formulation in more complex conditions (3D vascularized tumor model).

In conclusion, this set of experiments proved that the encapsulation process does not hinder virus infectivity and killing ability, whereas it increases the infectivity in a CAR-independent manner.

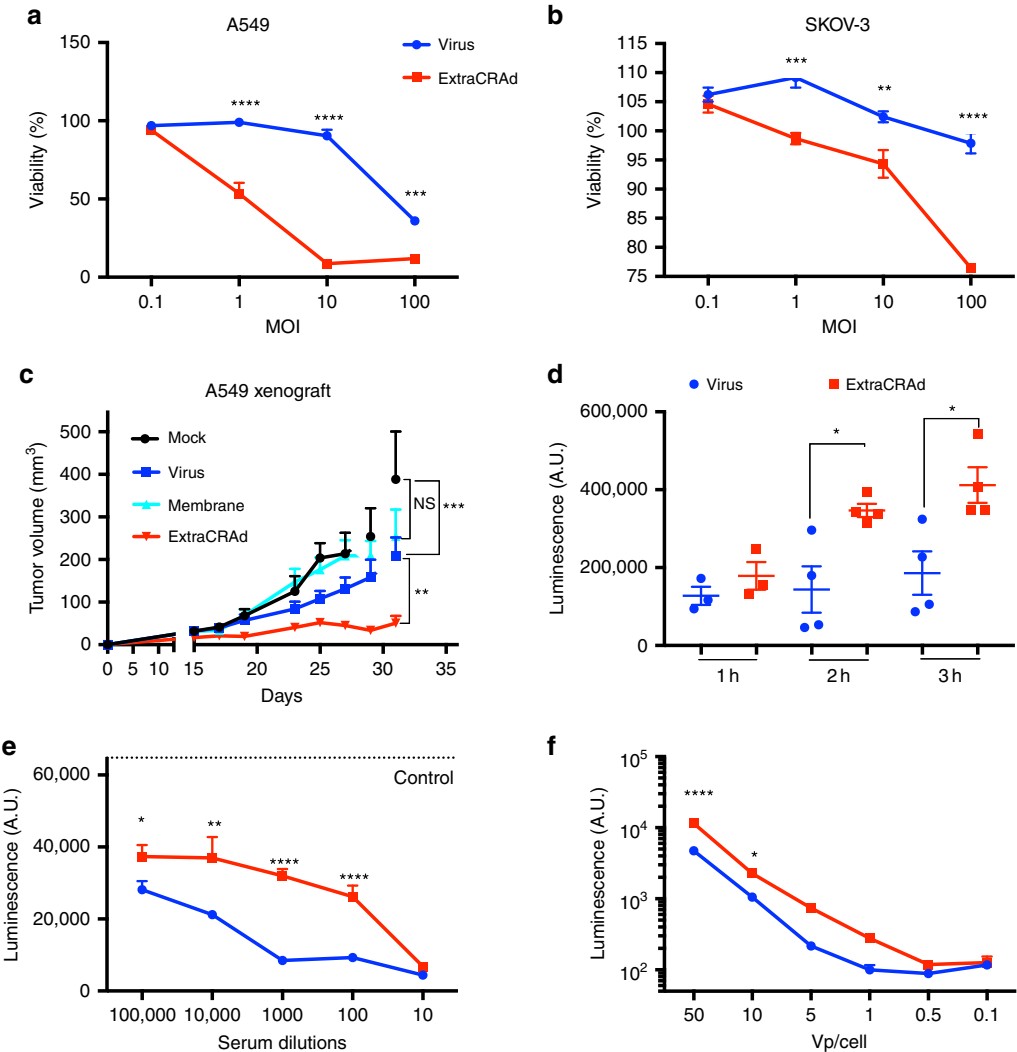

**Fig. 3 ExtraCRAd mechanism of infectivity, transduction, and antibody resistance.** Infectivity assay on **a** A549 (high-CAR) and **b** SKOV-3 (low-CAR) cell lines. Both the cell lines were infected with ExtraCRAd (red) or the naked virus (blue). Cell viability at 3 days was evaluated by the MTS assay. The data are presented as mean ± s.d. ($n = 3$). **c** In vivo assessment of the oncolytic efficacy of the naked virus compared to ExtraCRAd in A549 xenografts in nude mice. The mice were injected subcutaneously with $5 \times 10^6$ cells per flank. The treatments (mock, PBS; CCM, cell membrane vesicles derived from $3 \times 10^6$ A549 cells; Virus, $1 \times 10^9$ viral particles; ExtraCRAd, $1 \times 10^9$ viral particles co-extruded with $3 \times 10^6$ A549 cells) were injected at day 15 and at day 25 post tumor implantation. The data are presented as mean ± SEM ($n \geq 8$) and were analyzed by two-way ANOVA followed by Tukey's post-test. Error bars represents SEM. The levels of statistical significance were set at probabilities $**p < 0.01$ and $***p < 0.001$. **d** Uptake kinetic study: the differences in the uptake kinetic were assessed by infecting A549 cells with ExtraCRAd or naked virus (adenovirus modified to express luciferase) for 1, 2, or 3 h, and by 24 h incubation to allow the expression of luciferase. The data are presented as mean ± s.d. ($n \geq 3$) with the dot plot of the single replicates. Unpaired $t$-test was used to assess statistical significance. Neutralizing antibodies assay performed with **e** plasma from pre-immunized mice and **f** with anti-adenovirus serotype 5 antibody. ExtraCRAd and the naked virus were pre-incubated with the serum of pre-immunized mice at different dilutions (**e**) or with the antibody diluted 1:2 000 (**f**) for 60 min before the infection. A549 cells were infected with ExtraCRAd or the naked virus for 2 h, followed by 24 h of incubation, before being lysed. The data are presented as mean ± s.d. ($n \geq 3$). Analysis with a two-way ANOVA, followed by Fisher LSD post-test was performed in **f**, while a two-way ANOVA, followed by Sidak post-test, was used to assess statistical significance in all the experiments, unless otherwise specified. Levels of significance were set at $*p < 0.05$, $**p < 0.01$, $***p < 0.001$, and $****p < 0.0001$. Error bars represent s.d., unless otherwise specified.

**ExtraCRAd cellular internalization mechanisms**. One of the disadvantages of oncolytic adenoviruses is their need for the CAR receptor in order to infect cancer cells[18]. This may lead to loss of efficacy of the therapy due to downregulation of the CAR receptor by the tumor cells[19,20]. On the contrary, generally the uptake of nanoparticles is CAR receptor-independent and is mediated by different mechanisms according to the size, surface charge, shape, and presence of a protein corona[21–23]. The uptake mechanisms can be passive (diffusion of substances through the cytoplasmic membrane) or active (phagocytosis;

receptor-mediated endocytosis; micropinocytosis)[24]. Given the hybrid nature of ExtraCRAd, we sought to investigate its uptake mechanism. We evaluated if ExtraCRAd entered preferably through receptor or following the behavior of a lipid structure fusing with the cancer cell membrane. Firstly, we investigated the kinetic of the uptake (Fig. 3d) by infecting A549 cells with ExtraCRAd or with the naked virus for 1, 2, or 3 h before removing the sample in the supernatant. The cells were then incubated overnight to allow the expression of the luciferase transgene and lysed after 24 h, quantifying the luminescence

produced. ExtraCRAd displayed increased entry in the cells after 2 and 3 h infection when compared with the naked virus. Then, we elucidated the uptake mechanism (either active or passive) by evaluating the differences between ExtraCRAd and control virus at low temperature (on ice) (Supplementary Fig. 6a). No statistically significant difference was found between ExtraCRAd and the virus. Taking into account the size of the particles (around 100 nm), these results suggest an active uptake process and not a passive fusion of the system with the cell membrane[25,26]. Thereby, chlorpromazine and sucrose were chosen as suitable inhibitors to elucidate a difference in the uptake of virus or ExtraCRAd: both the compounds act on clathrin/caveolin mediated endocytosis[21]. The incubation with sucrose resulted in comparable reduction of the uptake between coated and uncoated virus (Supplementary Fig. 6b), while, for chlopromazine, the inhibition of the uptake was higher for ExtraCRAd (Supplementary Fig. 6c). These results suggest that ExtraCRAd relies on a clathrin-mediated endocytosis much more than the naked adenoviruses, due to the different size, composition, and surface properties of ExtraCRAd[26].

Another downside associated with the clinical use of oncolytic viruses is the high prevalence of neutralizing antibodies against several serotypes of adenoviruses in the human population[27,28]. We hypothesized that the coating of the viral capsid with cell membranes could shield the virus from neutralizing antibodies, as nanoparticles coated with red blood cell membrane have shown increased circulation time[29]. Firstly, we tested this hypothesis by studying the inhibition of ExtraCRAd infectivity in presence of serum from mice immunized with adenovirus (1 month prior to sacrifice by injecting subcutaneously $1 \times 10^9$ viral particles, once a week for 4 weeks). Serum was then isolated and used as source of adenovirus neutralizing antibodies. As shown in Fig. 3e, the extrusion process with cell membrane could only shield the virus at highest concentrations. To eliminate any influence by plasma proteins, we further investigated the shielding effect using an anti-hexon monoclonal antibody (Fig. 3f). ExtraCRAd effectively hid the virus from the monoclonal anti-hexon antibody, at the highest viral concentration tested, as testified by the increased expression of luciferase.

In conclusion, we demonstrated the independence of Extra-CRAd uptake from CAR receptors and the protective effect offered by the cell membrane towards neutralizing antibodies. These results showed an increased infectivity and efficacy of the virus in vitro when wrapped into the cancer cell membrane.

**ExtraCRAd controls tumor growth in different mouse tumors.** After the characterization of the structure of ExtraCRAd and its mode of action in vitro, we proceeded to assess the efficacy of ExtraCRAd in murine models of melanoma and lung cancer in vivo. To this end, B16.OVA, B16.F10, and LL/2 murine cancer cells were injected in the right flank of C57BL/6 mice to establish subcutaneous tumor models ($n = 8$ mice per group, one tumor per mouse). The treatment schedule was performed according to the following protocol: for B16.OVA, intratumoral injections ($n = 4$, injection per tumor) were performed every 2 days, starting from day 8 after the tumor inoculation; as for B16F10, given its aggressiveness and fast tumor growth, the first injection was performed on day 6 after tumor inoculation, keeping the same treatment schedule (4 injections administered every 2 days); for LL/2, the first injection was performed on day 8 after tumor implantation, following the same scheme (four injections every 2 days). The efficacy of ExtraCRAd in the treatment of the tumor models is presented in Fig. 4. The mean tumor growth curves for B16.OVA are shown in Fig. 4a, B16.F10 is reported in Fig. 4b and LL/2 in Fig. 4c. In all three the tumor models, the treatment

with ExtraCRAd significantly slowed down the progression of the tumors when compared to virus alone, virus mixed with membranes and ExtraCRAd wrapped with a mismatched cell membrane. The single tumor growth curves for the three models are presented in Supplementary Fig. 7 in supporting information. In the B16.OVA model, the controls (virus and membranes mixed, but not extruded, and membrane only) groups showed only 33% of responders, while in the more aggressive B16.F10 model the effect of mix and virus was even more limited (12.5% and 28%, respectively). As for the LL/2 model, the membrane and virus controls did not affect the tumor growth in any of the animals; the mix of membranes and virus could control the growth in 40% of the animals. However, these results highlight the potential efficacy of the single components of ExtraCRAd (virus as an adjuvant and tumor lysate as the source of antigens) in immunogenic tumors (melanoma) while only the mix of the two components was partially effective in the lung model. Strikingly, homologous ExtraCRAd treatment significantly controlled the tumor growth in all tumor models. In the B16.OVA model, all the mice treated with ExtraCRAd responded to the therapy significantly better than all the other groups. In B16.F10 model, 62.5% of the animals responded to the treatment. In LL/2 tumor model, homologous ExtraCRAd was able to control the tumor growth in all the animals treated, while ExtraCRAd formulated with a heterologous cell membrane derived from a mouse bladder cancer cell line MB49 could affect the tumor growth in only half of the animal cohort. Thereby, we showed that wrapping an oncolytic virus with a tumor cell membrane significantly enhances the virus efficacy in several tumor models. Most importantly, we proved that the highest efficacy is achieved when the virus is coated with homologous, tumor matched, cell membrane.

**ExtraCRAd mediates specific anti-tumor response in B16. OVA.** In order to elucidate the mechanism of action of Extra-CRAd, tumors and lymphoid organs from the experiments described above, were harvested to perform immunological analyses. As presented in Fig. 5a, the study of the immunological profile in the tumor microenvironment (B16.OVA model) showed an increased presence of dendritic cells cross-presenting the tumor-specific antigen SIINFEKL on their surface, resulting in an increase in the potential priming of the effector T cells. Moreover, the analysis of the tumor microenvironment highlighted also an increased number of OVA-specific CD8$^+$ T cells in the ExtraCRAd group compared to mock, virus, membrane, and mix groups (Fig. 5b). The majority of these cells were presenting PD-1 receptor expressed on their surface, which suggest the activation of the T-cell following the stimulation of T-cell receptors (Fig. 5c)[30]. However, OVA-specific T cells represented only a small portion of the experienced T cells, indicating that ExtraCRAd therapy could potentially elicit the activation of a variety of different T cells. In fact, the total pool of CD8$^+$ T cells showed a more antigen-experienced profile due to the upregulation of PD-1$^+$ on their surface (Fig. 5d) in mice treated with ExtraCRAd. Interestingly, we observed an increase in the antigen-experienced T cells also in the mix control group, highlighting the importance of the virus as adjuvant and the cell membrane vesicles as sources of the tumor antigens. As previously described in the work of Capasso et al.[12], a virus bearing cancer-specific peptide represents an efficient strategy to train the immune system in mounting a cancer-specific response employing an oncolytic virus as adjuvant. We implemented such technology by adopting the properties of a tumor lysate to reach a broader, but still cancer-specific effect. Tumor lysate elicits different beneficial mechanism in cancer patients by inducing rapid (24–48 h) and

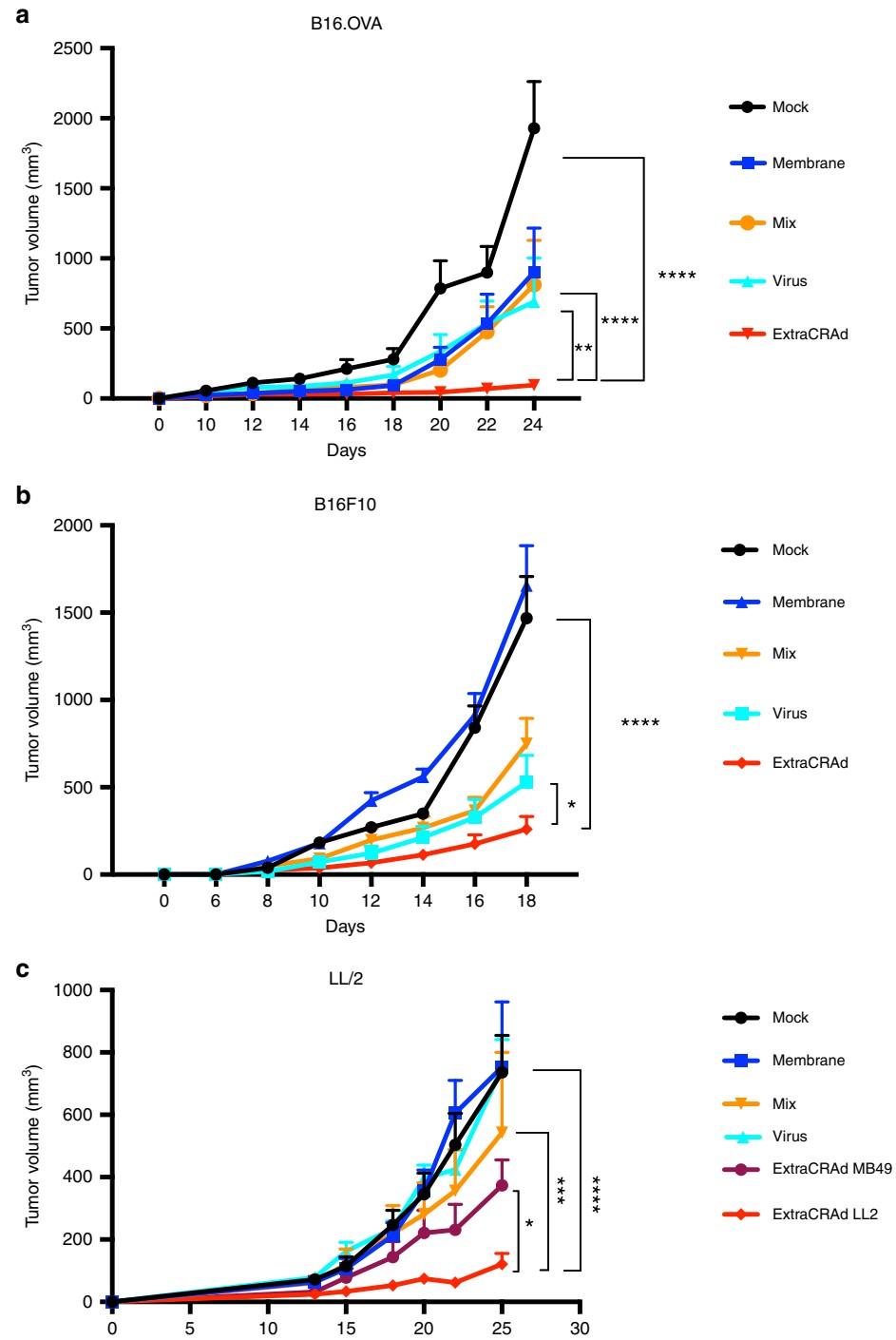

**Fig. 4 Efficacy of the treatment with ExtraCRAd in three mouse tumor models.** Median tumor growth curves for **a** B16.OVA, **b** B16.F10, and **c** LL/2. In the median curves the data are presented as mean ± s.d ($n \geq 7$). Error bars represent s.d. The results were analyzed with a two-way ANOVA, followed by Tukey's (**a**, **b**) and Dunnet's (**c**) post-test comparison, and the levels of significance were set at the probabilities of *$p < 0.05$, **$p < 0.01$, ***$p < 0.001$, ****$p < 0.0001$. Mice were injected every second day four times starting at day 8 post-injection for B16.OVA and LL/2 and at day 6 post-injection for B16.F10.

committed maturation to Th1/Th17 polarizing DCs[7] and eradicating tumor in murine models when combined with CpG oligonucleotides[31]. Dendritic cells are then able to prime multiple pools of T-cells favoring the Th1 lineage[32]. To capitalize on these previous findings, ExtraCRAd was designed to include tumor lysate, viral CpG and oncolytic properties in a single vaccine particle, therefore providing an enhanced version of cell-lysate-based vaccines. Importantly, our platform allows for co-delivery antigens and adjuvants into the same APC, which has been

shown to be critical in terms of inducing sufficient CD8 T-cell responses[33–36].

**ExtraCRAd increases DCs and T-cells in aggressive melanoma.** The tumors and spleens belonging to mice inoculated with B16.F10 cells were collected, processed into single-cells suspension, and labelled with anti-(a)CD8 for T-cells, aCD19 for B-cells, gp100 and TRP-2 pentamers for tumor-specific T cells, aPD1 for

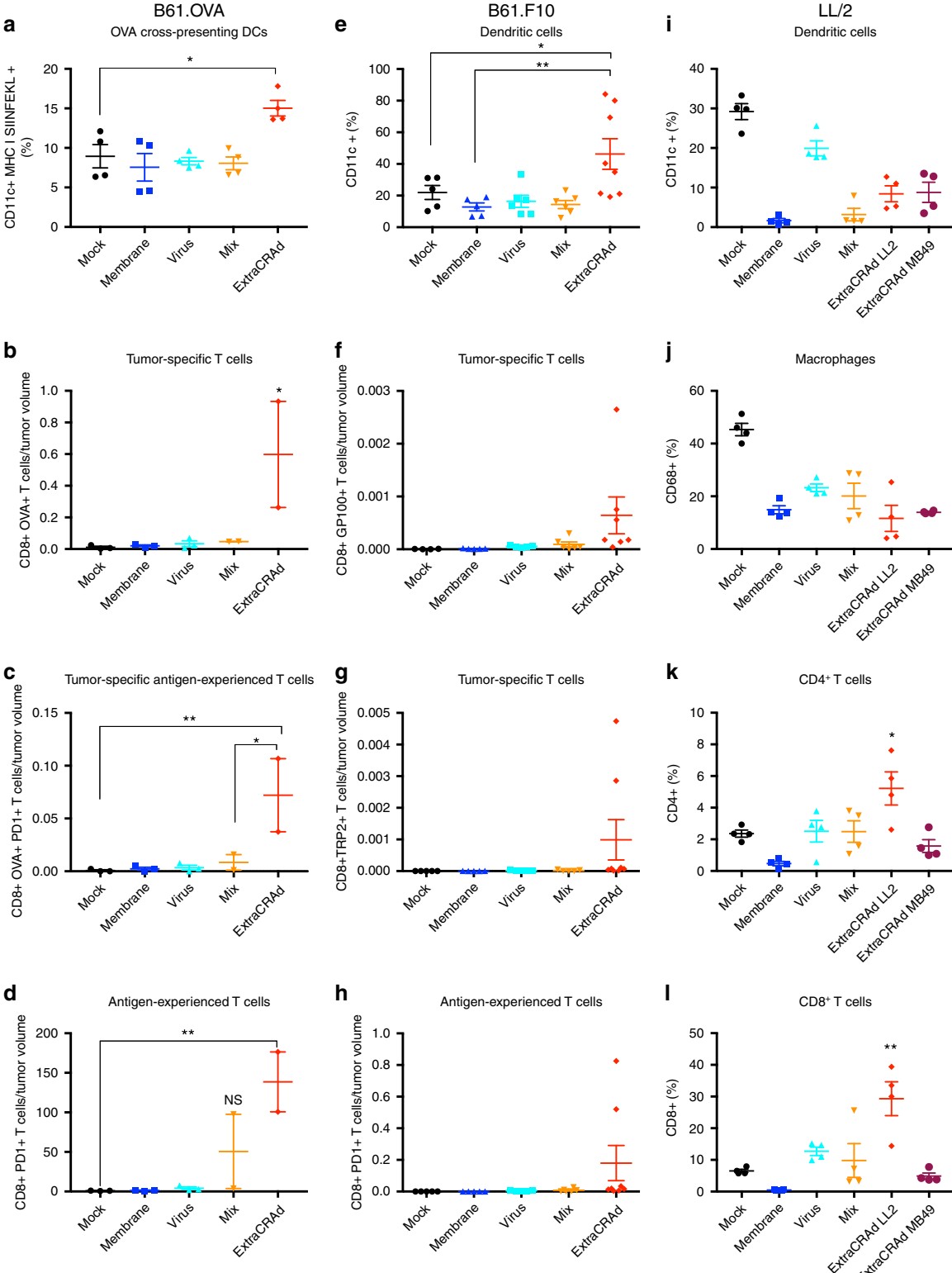

antigen-experienced T-cells, aCD68 for macrophages identification, aCD11c for dendritic cells, and aCD370 for cross-presenting activated dendritic cells.

Tumors treated with ExtraCRAd showed an increase in the number of tumor-specific T cells, as in the number of PD-1+ T cells (Fig. 5f–h). All these results confirmed our previous findings in the B16.OVA model (Fig. 5a–d). We also recorded an increase in the number of dendritic cells (Fig. 5e), macrophages

(Supplementary Fig. 8a) and total CD8+ T cells (Supplementary Fig. 8b) within the tumor microenvironment, showing a complex and general interaction of different immunological effector cells. Interestingly, the amount of CD8+ and CD11c+ cells increased in the spleens of mice treated with ExtraCRAd (Supplementary Fig. 9a, c). A further investigation with CD370 marker revealed an augmented cross-presentation activity of the DCs in the spleen (Supplementary Fig. 9b) and in the number of tumor-specific

**Fig. 5 Immunological analyses of the tumor microenvironment.** B16.OVA **a** Percentage of dendritic cells cross-presenting OVA (SIINFEKL) in the context of MHC class I in the tumor microenvironment, **b** OVA-specific, **c** OVA-specific antigen-experienced, and **d** antigen-experienced T cells in the tumor microenvironment. The number of T cells was normalized by the tumor volume. Mix identifies a treatment composed of cell membrane vesicles just mixed with adenovirus without extrusion. The results are presented as mean ± s.d ($n \geq 3$) and the dot plot of the single replicates. The data were analyzed with a two-way ANOVA, followed by Dunnet (**a**–**c**) or Tukey (**d**) post-test comparison. The levels of significance were set at the probabilities of *$p < 0.05$, **$p < 0.01$, ***$p < 0.001$, ****$p < 0.0001$. B16F10 **e** CD11c$^+$ dendritic cells percentage in the tumor microenvironment and **f** gp100 and **g** TRP2 pentamer-specific T-cells, identifying the CD8$^+$ tumor-specific T-cells in the tumor microenvironment. **h** Antigen-experienced T-cells measured by the expression of PD-1 on T-cells. Mix identifies a treatment composed of cell membrane vesicles just mixed with adenovirus without extrusion. The data are presented as mean ± s.d ($n \geq 7$), together with the dot plot of the single value. The data were analyzed with two-way ANOVA, followed by Fisher LSD post-test. The levels of significance were set at the probabilities of *$p < 0.05$, **$p < 0.01$. LL/2 **i** Percentage of CD11c$^+$ dendritic cells in the tumor microenvironment in the LL/2 model. **j** Percentage of CD68$^+$ macrophages. **k** Percentage of CD4$^+$ and **l** CD8$^+$ T-cells in the tumor microenvironment. Mix identifies a treatment composed of cell membrane vesicles just mixed with adenovirus without extrusion. The data are presented as mean ± s.d. ($n = 4$) and were analyzed by two-way ANOVA, followed by Dunnet's post-test. The levels of significance were set at the probabilities of *$p < 0.05$ and **$p < 0.01$. The error bars represent s.d.

T cells (Supplementary Fig. 9d). Thus, we hypothesize that ExtraCRAd elicits a broader response, including tumor-specific and non-specific immune response[37].

**ExtraCRAd primes a strong immunity in a solid tumor model.** The efficacy of immunotherapies is greater in loose, permeable, tumors (e.g., melanoma) when compared to solid tumors due to the physical hindrance to the migration of immune cells within the tumor microenvironment[38–40]. Thereby, after having evaluated the efficacy of ExtraCRAd into a solid tumor model (LL/2 lung cancer), we investigated the immune contexture in the tumor microenvironment, spleen, and draining lymph nodes. As reported in Fig. 5i, j, the treatment with ExtraCRAd does not result into an increase in the fraction of dendritic cells or macrophages in the tumor microenvironment. However, the tumors treated with homologous ExtraCRAd are infiltrated by a statistically significant percentage of both CD4$^+$ and CD8$^+$ T-cells (Fig. 5k, l). These changes are associated with a local immune response as suggested by the immunological profile of the spleen (Supplementary Fig. 10), which displays no differences between the treatments, compared to the immune contexture of the draining lymph node (Supplementary Fig. 11). In particular, the draining lymph nodes of animals treated with homologous ExtraCRAd present an increased population of APCs (Supplementary Fig. 10a), together with a significant increase in the CD8$^+$ T-cell population (Supplementary Fig. 10c). These results suggest that ExtraCRAd can induce anti-tumor immunity by simultaneously providing both antigenic material and immunostimulatory signal to DCs, both of which are needed for efficient T-cell priming to occur in the draining lymph nodes.

**ExtraCRAd efficacy as preventive vaccination.** Lastly, we investigated whether a preventive vaccination scheme with ExtraCRAd would protect against tumor challenge, control the tumor growth and affect the overall long-term survival. The pre-immunization set-up allows to evaluate the efficacy of the formulation in creating a memory immune response without confounding factors, including the release of Damage-associated molecular pattern (DAMPs) in the tumor microenvironment following each administration of the therapy[41]. To this end, we vaccinated the animals for a total of three times before challenging them with CMT64.OVA or B16F10 tumor cells. As reported in Fig. 6a, b, the immunization with ExtraCRAd wrapped with homologous tumor-matching membranes (ExtraCRAd CMT64.OVA in the CMT64.OVA model and ExtraCRAd B16.F10 in the B16.F10 model) prolongs the overall survival on the animals with >50% of the animals still alive at day 40 after tumor engraftment (CMT64.OVA) and at day 28 after tumor implantation in the

B16.F10 model. Both the control of naked virus and ExtraCRAd wrapped in a heterologous miss-matched tumor membrane do not affect the overall survival, with none (CMT64.OVA) or one animal (B16F10) still alive at the end of the experiment.

The survival data is reflected in the tumor growth graphs, where homologous ExtraCRAd was able to significantly inhibit tumor growth compared to other treatment groups, as presented in Fig. 6c, d. In the CMT64.OVA model, the vaccination with naked virus or miss-matched ExtraCRAd does not have a significant effect on the tumor growth, while the homologous ExtraCRAd statistically slowed the tumor growth rate. More interestingly, in the aggressive melanoma model B16.F10 we observed a partial control over the tumor growth mediated by the vaccination with naked virus and with the heterologous ExtraCRAd compared to the mock group. Nevertheless, homologous ExtraCRAd vaccination was found to be the most effective regimen in these tumor challenge models. Overall, these results indicate that a vaccination scheme with ExtraCRAd wrapped in a matched membrane is required for induction of tumor-specific immunity and for the therapeutic and/or preventive efficacy, leading to sustained tumor growth control.

## Discussion
The recent discoveries in cancer immunotherapy contributed to a shift in the paradigm of cancer treatment from a preconditioning wiping out the immune system before the administration of chemotherapeutics to the exploiting of the immune system's ability for the precise targeting and killing of cancer cells[42]. The application of immune check-point inhibitors in the clinics contributed to this revolution and, at the same time, highlighted the limits of cancer immunotherapy[43]. There is a urgent need for the discovery and delivery of tumor neoantigens to prime more efficiently the patient's immune system. A recent compromise suggests the use of membranes derived from cancer cells as crude sources of neoantigens, especially in the case of highly mutagenic tumors. The isolation and further processing of this biological material are relatively easy making them suitable for applications in the development of cancer vaccines[44].

In parallel, there is a high demand for potent and safe cancer vaccines able to effectively stimulate antigen presenting cells and break the tumor-induced immunotolerance[45]. The development of virus nanoparticles carrying cancer moieties represent a very versatile technology with wide applicability to different kind of cancers to stimulate an immune response. Virus-like particles have been widely employed as adjuvants in the formulation of cancer vaccines[46], while oncolytic virus are currently being re-evaluated as powerful vaccines shaped by millennia of co-existence with the immune system, with less focus on their oncolytic efficacy[47].

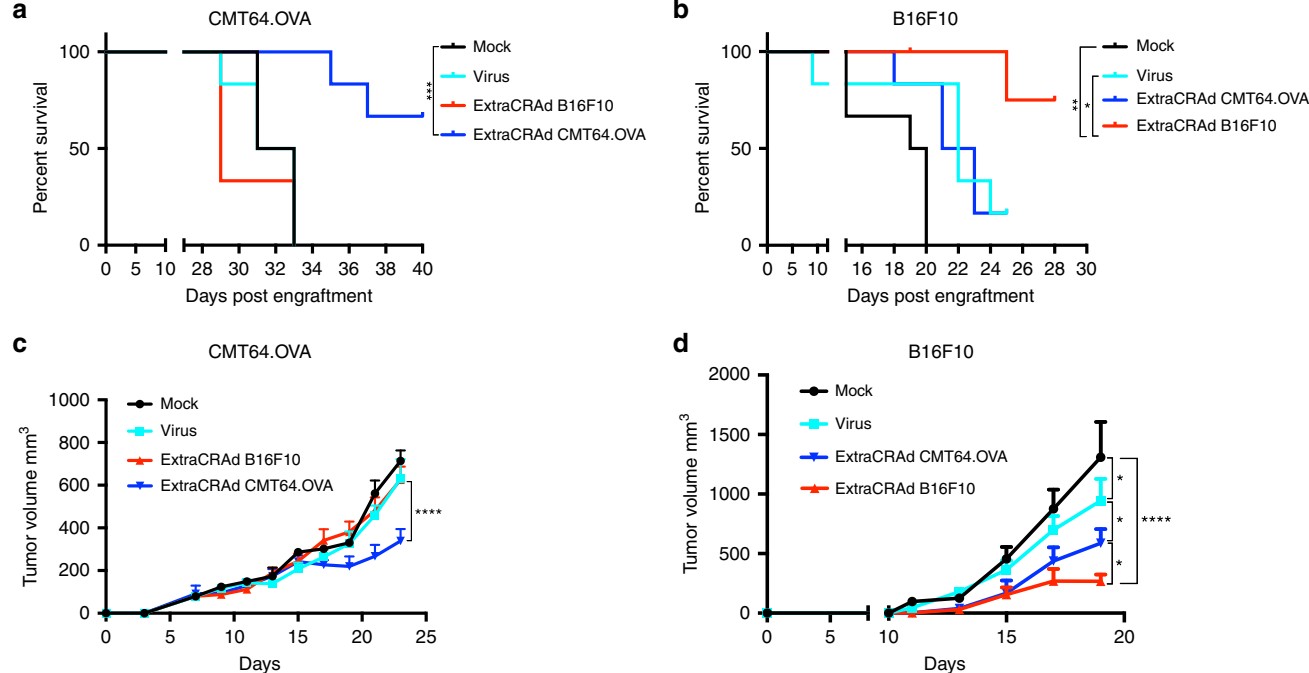

**Fig. 6 Efficacy of ExtraCRAd in a pre-immunization set-up in melanoma and lung cancer models. a** Long-term overall survival of C57BL/6 mice injected with $7 \times 10^6$ CMT64.OVA subcutaneously in the flank. **b** Long-term overall survival of C57BL/6 mice injected with $1 \times 10^5$ B16.F10 cells subcutaneously in the flank. In both the models, animals were removed from the survival curve upon sacrifice. Mice dead for causes not imputable to the tumor growth or mice still alive at the completion of the study were censored in the survival curves. The data were analyzed with log-rank (Mantel–Cox) curve comparison. The levels of significance were set at \*$p < 0.05$, \*\*$p < 0.01$, and \*\*\* $p < 0.001$. Mean tumor growth curve (mm³) in **c** the CMT64.OVA model and **d** the B16. F10 model. The data are presented as mean ± SEM ($n = 6$). The data were analyzed with two-way ANOVA followed by Tukey's (**c**) and Fisher's (**d**) post test. The levels of significance were set at probabilities of \*$p < 0.05$ and \*\*\*\*$p < 0.0001$. The error bars represent SEM.

In this work, we develop a method to coat an oncolytic virus with a cancer cell membrane (ExtraCRAd) and we provide evidence of its efficacy in regulating the tumor growth in different murine cancer models. We prove how the coating of a virus by extruding a cancer cell membrane on its surface is beneficial and exhibits synergistic effects of its components, carrying a wider spectrum of activity as cancer vaccine. Compared to other type of cancer vaccinations, such as peptides vaccine or simply the use of naked oncolytic virus, ExtraCRAd brings the advantage to combine in the same particle the adjuvant (virus) and the antigens (tumor-associated proteins). This might lead to a more pronounced immune response without the need of necessarily knowing the associated tumor antigens. We observe advantages over the single components both in vitro and in vivo. The viral transfection is significantly increased when the virus is wrapped in the cancer membrane, implying an uptake mechanism different from the CAR-mediated one of the naked virus and resulting in increased viral infection also for CAR-negative cell lines. This finding is indeed well in line also with other approaches where adenovectors have been incapsulated in liposomes[48–50]. In a more clinically relevant tumor model, constituted by high-CAR human cancer cells implanted into nude mice, ExtraCRAd outperformed the naked virus in controlling the tumor growth with a purely oncolytic effect, suggesting again that the entry of the wrapped virus does not only depends on the expression of the CAR receptor but it can use different mechanisms that might increase the efficacy of the virus in absence of the immune system. Furthermore, the technology contributed to partially shield the virus from neutralizing antibodies. This is a very interesting observation that might open up a different use and route of administration for this nanovaccine rather than intratumoral administration. In fact, neutralizing antibodies have been a difficult obstacle to overcome in classical gene therapy approaches where viral vectors are delivered intravenously and are expected to target a specific tissue[51–53]. In the context of oncolytic viruses, and even more in oncolytic cancer vaccines, the interest for neutralizing antibodies has diminished as very often these vaccines are almost exclusively given by intratumoral administration[54,55]. Our approach shows potential in eliciting the adaptive immune system increasing the number of both macrophages and dendritic cells, most of them activated to present tumor antigens, and later, cancer-specific repertoire to fight the cancer itself. This is in line with what observed with similar approaches where similar immunological endpoint have been observed[12,56,57]. In addition and differently from other works, we proved the efficacy of a preventive vaccination scheme with tumor-matched ExtraCRAd in prolonging the overall survival and in controlling the tumor growth in pre-immunization experiments. The tumor type matched membrane viral particle shows longer survival and efficacy in slowing down the tumor growth while the non-matched one does not provide a general advantage in protecting against the tumor. These results indicate that the vaccination efficacy is not dependent on the technology but on the need for specific tumor-matched antigens over ExtraCRAd. Overall, our results suggest that ExtraCRAd is a versatile and advanced platform for cancer treatment with an interesting potential for present and future clinical impact given its easy tailorability to each patient, choosing a suitable virus and obtaining cancer cells from biopsy. More investigations to assess the efficacy of this technology in additional tumor models and clinical settings are needed.

## Methods
**Isolation of the cell membrane**. We isolated cell membranes from B16.OVA and B16.F10 murine melanoma cells, LL/2 and CMT64.OVA murine lung cancer, MB49 murine bladder cancer, human A549 lung cancer, and SKOV-3 ovarian

cancer cells, according to the protocol previously published[14,58]. B16.OVA cells were cultured in 10% heat inactivated fetal bovine serum (FBS) in Roswell Park Memorial Institute (RPMI)−1640 medium supplemented with 1% penicillin-streptomycin, 1% non-essential amino acids, and 1% L-glutamine. In order to select the cells positive for OVA, geneticin (G418 Sulfate, Thermo Fisher, USA) was added to the medium, at a concentration of 5 mg/ml. All the other cell lines were cultured according to American Type Culture Collection protocols. In order to retrieve the membrane, the cells were washed with $1 \times$ phosphate buffer solution–ethylenediaminetetraacetic acid (PBS–EDTA; pH 7.4) solution, and detached. The cells were centrifuged at $660 \times g$ for 10 min and washed three times with $1 \times$ PBS (pH 7.4). The cell pellet was resuspended into lysing buffer (20 mM of TRIS HCl; Sigma-Aldrich, USA; 10 mM of KCl; Sigma-Aldrich, USA; 2 mM of MgCl$_2$; Sigma-Aldrich, USA; 1 protease inhibitor mini tablet, EDTA free; Pierce, Thermo Fisher, USA) and pipetted thoroughly. We centrifuged the cells at $3200 \times g$ for 5 min, collected the supernatant, and repeated the procedure, centrifuging the cells a second time at $3200 \times g$ for 6 min. We pooled the supernatant and centrifuged it at $21,000 \times g$ for 25 min at $+4\,°C$. We then collected the supernatant and centrifuged it at $45,000 \times g$ for 5 min in a TLA 120.0 rotor in an ultracentrifuge (Optima Max, Beckmann Coulter, USA) at $+4\,°C$. The supernatant was then discarded, and we resuspended the membranes in $1 \times$ PBS prior to extrusion.

**Encapsulation of Ad5D24-CpG virus within cell membrane.** ExtraCRAd was prepared using Ad5-D24-CpG virus together with cell membrane fragments by extrusion through a polymeric membrane (0.8 μm, Nucleopore Track-Etch Membrane, Whatman, UK) in an extruder (Avanti Polar Lipids, USA). The virus and the membranes were resuspended in $1 \times$ PBS solution and extruded 5, 10, 20, 30 times through the membrane. For the final formulation, 20 passages were selected as optimal conditions for the complete encapsulation of the virus within cell membrane vesicles.

**Nano-tracking analyses.** Extruded virus, cancer membrane and ExtraCRAd were analyzed using Nanosight model LM14 (Nanosight) equipped with blue (404 nm, 70 mW) laser and SCMOS camera. The samples were diluted in DPBS and three 60 s videos were recorded using camera level 13. The data was analyzed using NTA software 3.0 with the detection threshold 5 and screen gain at 10 to track as many particles as possible with minimal background.

**Cryo-transmission electron microscope.** About 3 μl of fresh samples were snap frozen on a carbon-coated copper grid and imaged with JEOL JEM-3200FSC TEM, with 300 kV field emission at different magnifications.

**Cell lines.** The human lung carcinoma cell line A549, human ovarian adenocarcinoma SKOV-3, the mouse melanoma cell line B16.F10, the mouse LL/2 lung cancer line and the mouse bladder cancer cell line MB49 were purchased from the American Type Culture Collection (ATCC; Manassas, VA, USA). The cell line B16. OVA, a mouse melanoma cell line expressing chicken OVA, was kindly provided by Prof. Richard Vile (Mayo Clinic, Rochester, MN, USA). The lung adenocarcinoma cell line CMT64.OVA was a kind gift from Florian Kuhnel (Hannover, Germany). All cell lines were cultured under appropriate conditions and were routinely tested for mycoplasma contamination.

**Preparation of conditionally replicating adenoviruses.** All CRAds were generated, propagated, and characterized using standard protocols, as previously described[59]. All viruses used in this study have been previously reported: Ad5D24 is an adenovirus that features a 24-base-pair deletion (Δ24) in the E1A gene, Ad5 Δ24-CpG is a CRAd bearing a CpG-enriched genome in the E3 gene[60]. Ad5-luc is a non-replicating adenovirus carrying luciferase transgene[61].

**Zeta (ζ)-potential and dynamic light scattering analysis.** Samples were prepared as described in the previous section. Each sample was then vortexed and diluted to a final volume of 700 ml with sterile milli-Q water adjusted to pH 7.4, after which the sample was transferred to a polystyrene disposable cuvette to determine the size of the complexes. Afterward, the sample was recovered from the cuvette and transferred to a DTS1070 disposable capillary cell (Malvern, Worcestershire, UK) for zeta potential measurements. All measurements were performed at 25 °C with a Zetasizer Nano ZS (Malvern).

**Cell viability assay.** MTS assay was performed according to the manufacturer's protocol (CellTiter 96 AQueous One Solution Cell Proliferation Assay; Promega, Nacka, Sweden). Spectrophotometric data were acquired with Varioskan LUX Multimode Reader (Thermo Scientific, Carlsbad, CA, USA) operated by SkanIt software. About 10,000 cells were plated in 96-well plate 1-day prior infections. Three days post-infection cell viability was measured using the CellTiter-Fluor™ Cell Viability Assay (Promega), and a multi-well plate reader (Varioskan LUX; ThermoLabsystems) was used to determine the fluorescence of the samples.

**Transfection assay.** A549 cells were infected with Ad5-Luc for 1, 2, and 3 h in 2% FBS in low-glucose Dulbecco's modified Eagle's medium (DMEM) medium. After the infection time medium was discarded, cells washed twice with PBS and new 10% FBS in low-glucose DMEM added. After incubation overnight, cells were lysed with NP40 lysis buffer and luciferase level analyzed by plate reader Varioskan LUX.

**Neutralizing antibody assay.** Mice were immunized by a single subcutaneous injection of $1 \times 10^9$ viral particles of Adenovirus5 Δ24-CpG. One month later, animals were sacrificed and blood was collected through heart puncture. After 15 min at room temperature blood cloth was removed and vials centrifuged at 4 °C for serum separation. Serum was then collected and stored at −20 °C. For the NAb assay, Ad5-LUC virus was incubated for 1 h at 37 °C with serum and further used to infect cells for 2 h. In vitro assay was performed with monoclonal antibody against viral hexon protein (Novus Biological, Littleton, CO, USA). Luciferase expression levels were detected by Varioskan LUX.

**Sample preparation for proteomics.** Three parallel cell membrane isolations followed by extrusion and/or encapsulation, were performed as described above using human lung cancer A549. Samples were solubilized in 0.2% of RapiGest SF (Waters Inc., Milford, MA, USA), dried in a speed vacuum (Eppendorf Concentrator Plus; Eppendorf AG) and resuspended in 50 mM of ammonium bicarbonate buffer, pH 7.8 (AMBIC). The protein concentration was determined with a standard BCA protein assay kit (Thermo Fisher Scientific, Waltham, MA, USA). Samples were either diluted further or pipetted directly to a final volume of 50 μl in 50 mM of AMBIC, such that the final protein amount used for preparation of tryptic peptides was 7 μg. Tryptic peptides were prepared using Thermo Fisher Scientific's *In-Solution Tryptic Digestion and Guanidination Kit* according to the instructions of the manufacturer, but without the guanidination step. After overnight digestions, formic acid was added to a final concentration of 0.1%, incubated at 37 °C for 45 min, followed by centrifugation at $16,200 \times g$ for 15 min in order to remove Rapigest SF and particulate debris from the samples.

**Liquid chromatography–tandem mass spectrometry.** Digested peptides were put in auto sampler vials and loaded into an Easy-nLC 1200 (Thermo Fisher Scientific) coupled to an Orbitrap Fusion MS (Thermo Fisher Scientific). Chromatographic separation was carried out in commercially packed C18 columns (Acclaim PepMap 2 mm, 100 Å, 75 mm, 15 cm; Thermo Fisher Scientific). Peptides were loaded in buffer A (5% of acetonitrile and 0.1% of formic acid) and eluted with a 1 h linear gradient from 5 to 30% of buffer B (80% of acetonitrile and 0.1% of formic acid). Three biological replicates were sequentially injected with two 15-min wash runs and a 1 h blank run alternated between distinct 'treatments'. Mass spectra were acquired using a Top20 data-dependent method with an automatic switch between full MS and MS/MS (MS2) scans. The Orbitrap analyzer parameters for the full MS scan were resolution of 120,000 mass range of 350 to 1800 m/z, and AGC target of 4e5 ions, whereas those for MS2 spectra acquisition were resolution of 30,000, AGC target of 5e4 ions, an isolation window of 2 m/z and dynamic exclusion of 30 s. Column chromatographic performance was routinely monitored with intermittent injections of 50 fmol of a commercially available Bovine serum albumin (BSA) peptide mix (Waters Inc.), as well as evaluating double-wash runs for carryover peptides.

**Data processing.** Protein groups identification and quantification were carried out within the MaxQuant software, package, v. 1.6.1.0[62], with a UniProtKB human FASTA file containing 86,725 entries to which 245 commonly observed contaminants and all reverse sequences were added.

**Data analysis.** Enrichment analysis and hierarchical clustering were carried out with Perseus data analysis software v.1.5.6.0[63]. Abundance values were *log2* transformed, and protein identifications classified as being only identified by site, reverse sequences and potential contaminants were filtered out from the main data frame. Additionally, only identifications with non-zero intensity values in all three biological replicates from at least one 'treatment' were retained for comparisons. An intensity value that was less than the lowest intensity value from the entire data matrix was assigned to the proteins with missing quantification values to enable differential enrichment comparisons via hierarchical clustering.

**Pathway analysis.** Functional analyses were carried out within the gene ontology (GO) database using the enrichment analysis tool[64].

**Animal experiments and ethical permits.** All animal experiments were reviewed and approved by the Experimental Animal Committee of the University of Helsinki and the Provincial Government of Southern Finland. Female 4–6-weeks-old, C57BL/6JOlaHsd mice were obtained by ENVIGO. Female nude mice (HsdCpb: NMRI.Foxn1nu) 4–6-weeks-old were obtained by ENVIGO.

In the in vivo investigation of the oncolytic efficacy of ExtraCRAd, human lung cancer grafts were established into nude mice by injecting $5 \times 10^6$ subcutaneously in both the flanks (two tumors/mouse). The established tumors were treated with $1 \times 10^9$ virus particles twice, 15 and 25 day post implantation.

In the therapeutic vaccination set-up, tumors were established injecting $3 \times 10^5$ of B16-OVA murine melanoma cells, $1 \times 10^5$ of B16F10 murine melanoma cells, or $1.5 \times 10^5$ LL/2 murine lung cancer cells subcutaneously into the right flank of C57BL/6 mice. $1 \times 10^9$ virus particles were injected intratumorally four times every second day in established tumors starting at day 8 post implantation. At day of sacrifice tumor, spleen, and lymph nodes were collected from each mouse and frozen in −80 °C, in freezing media (10% of dimethyl sulfoxide media) for further immunological analyses.

In the pre-immunization set-up, C57BL/6 mice were immunized three times at days 1, 3, 14 with the equivalent of $1 \times 10^9$ virus particles in all the treatments. The mice were then injected on day 21 with $7 \times 10^6$ CMT64.OVA cells or $1 \times 10^5$ B16. F10 cells subcutaneously in the right flank. The tumor growth was measured every 2 days.

**Flow cytometry analysis**. Flow cytometry analysis was performed using a BD Accuri 6 plus (BD Biosciences) and analyzed by FlowJo software (Tree Star, Ashland, OR, USA). Epitope- specific T cells were studied using MHC Class I Pentamers (F093-84C-E, ProImmune, Oxford, UK). Other antibodies used included the following: murine Fc block CD16/32 (101320, Biolegend); FITC anti-mouse CD8 (A5402-3bE, ProImmune); PE/Cy7 anti-mouse CD19 (115520, Biolegend), FITC anti-mouse CD11c (117306, Biolegend), APC anti-mouse H-2Kb bound to SIINFEKL(116619, Biolegend), PE anti-mouse CD370 (143504, Biolegend), PE anti-mouse PD-1 (135206, Biolegend), PE anti-mouse CD68 (137013, Biolegend), APC anti-mouse CD4 (100412, Biolegend). All staining procedures were performed according to the manufacturer's recommendations. Gating strategies are shown in supplementary Fig. 12.

**Statistical analyses and correlation models**. Statistical analysis was performed using GraphPad Prism 7 (GraphPad Software, Inc., La Jolla, CA, USA). A detailed description of the statistical methods used to analyze the data from each experiment can be found in each figure caption.

**Reporting summary**. Further information on research design is available in the Nature Research Reporting Summary linked to this article.

## Data availability

All the data supporting the findings of this study are available within the article and its supplementary information files and from the corresponding author upon reasonable request. A reporting summary for this article is available as a Supplementary Information Files.

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

## Acknowledgements

We thank the Extracellular Vesicle Core Facility at the University of Helsinki, Finland, for providing the NTA service. We acknowledge the provision of facilities and technical support by Aalto University at OtaNano— Nanomicroscopy Center (Aalto-NMC). We thank Smart Servier Medical Art for the contribution to Fig. 1 (https://smart.servier.com). M.F. and C.C. thank the Doctoral Program in Drug Research for doctoral scholarships at the Faculty of Pharmacy, Helsinki University. H.A.S. acknowledges financial support from the HiLIFE Research Funds, the Sigrid Jusélius Foundation, the Academy of Finland (grant no. 317042), and the European Research Council Proof-of-Concept Research Grant (grant no. 825020). V.C. acknowledges the European Research Council under the Horizon 2020 framework (https://erc.europa.eu), ERC-consolidator Grant (Agreement no. 681219), Jane and Aatos Erkko Foundation (Project no. 4705796), HiLIFE Fellow (project no. 797011004) Cancer Finnish Foundation (project No. 4706116), Magnus Ehrnrooth Foundation (project No. 4706235).

## Author contributions

M.F., F.F., S.T., H.A.S., and V.C. conceived and planned the experiments. M.F. and F.F. carried out most of the experiments. C.C., S.F., B.M., J.C., K.P., L.Y., E.Y., F.H. helped in carrying out the experiments. O.K.K, J.N., H.A., and A.U. analyzed the samples in mass spectrometry and discussed the related results. J.T.H., H.A.S., and V.C. supervised the project. M.F., F.F., S.T., H.A.S., and V.C. wrote and corrected the paper. All authors provided critical feedback and helped shape the research, analysis, and manuscript.

## Competing interests

The authors declare no competing interests.
