## [Peer Review File · Nature Communications]

Reviewers' comments:

Reviewer #1 (Remarks to the Author):

The authors have addresses all of my previous concerns and substantially improved the manuscript. The data suffer from the limitations of testing a human-directed oncolytic virus in a murine immunocompetent model; however, there is nothing more they can do to address those limitations.

I only have minor comments that should be addressed:

1. In the abstract, the authors state “Virus-based cancer vaccines are nowadays considered among the most promising approaches in the field of cancer immunotherapy...”. I’m not aware of many positive clinical results with recombinant virus-based cancer vaccines, so I think this phrase needs to be reconsidered. Likewise, they state that “whole tumor lysate vaccines and oncolytic adenoviruses hold great promises for the future” in the introduction – I also don’t believe that the clinical data support this statement.

2. Panel 3F should use a logarithmic Y-axis to make it easier to see the data at low VP/cell.

Reviewer #2 (Remarks to the Author):

The authors have done a commendable job of addressing the comments from the previous round of review. They have provided thoughtful responses and clarifications, which are much appreciated.

An issue remains with the response to the query regarding the neutralizing effect of serum obtained from mice exposed to the virus (Reviewer 2, Question 3e). In theory, the membrane coating around

the virus should preclude drops in infectivity caused by neutralizing antibodies, whereas the data clearly shows that these antibodies are able to neutralize the coated ExtraCRAd formulation. As noted by the Reviewer, it cannot be claimed that 1:10 is very concentrated, particularly given that whole serum (as would be encountered in a patient) is 10 times more concentrated. While this may not affect the authors' in vivo studies in the present work, this does raise concerns about the real-world applicability of the platform. Does this mean that the treatment would only be effective in patients naïve to the virus? More discussion should be included along these lines.

**Artificially cloaked viral nanovaccine**

Manlio Fusiello[†], Flavia Fontana[†], Siri Tähtinen, Cristian Capasso, Sara Feola, Beatriz Martins, Jacopo Chiaro, Firas Hamdan, Otto K. Kari, Joseph Ndika, Erko Ylösmäki, Karita Peltonen, Leena Ylösmäki, Harri Alenius, Arto Urtti, Jouni T. Hirvonen, Hélder A. Santos, Vincenzo Cerullo**

REPLY TO REVIEWER'S COMMENTS

Reviewer #1:

The authors have addressed all of my previous concerns and substantially improved the manuscript. The data suffer from the limitations of testing a human-directed oncolytic virus in a murine immunocompetent model; however, there is nothing more they can do to address those limitations.

Re: We are grateful to the reviewer for the time spent reading our revised manuscript and for the valuable comments that were provided in all the revision stages.

I have only minor comments that should be addressed:

1. In the abstract, the authors state “Virus-based cancer vaccines are nowadays considered among the most promising approaches in the field of cancer immunotherapy...”. I am not aware of many positive clinical results with recombinant virus-based cancer vaccines, so I think this phrase needs to be reconsidered. Likewise, they state that “whole tumor lysate vaccines and oncolytic adenoviruses hold great promises for the future” in the introduction – I also don’t believe that the clinical data support this statement.

Re: We thank the reviewer for the comment. We modified the statement in the abstract as follows: “*Virus-based cancer vaccines are nowadays considered an interesting approach in the field of cancer immunotherapy...*”. We also modified the sentence in the introduction as follows: “*Amongst*

**Artificially cloaked viral nanovaccine**

Manlio Fusciello[†], Flavia Fontana[†], Siri Tähtinen, Cristian Capasso, Sara Feola, Beatriz Martins, Jacopo Chiaro, Firas Hamdan, Otto K. Kari, Joseph Ndika, Erko Ylösmäki, Karita Peltonen, Leena Ylösmäki, Harri Alenius, Arto Urtti, Jouni T. Hirvonen, Hélder A. Santos, Vincenzo Cerullo**

the different experimental treatments, active immunotherapy and, more specifically, viral therapy, hold great promises for the future”.

- 2. Panel 3F should use a logarithmic Y-axis to make it easier to see the data at low VP/cell.**

Re: We are grateful to the reviewer for the comment. We modified Figure 3F, changing the y-axis to logarithmic.

**Artificially cloaked viral nanovaccine**

Manlio Fusciello[†], Flavia Fontana[†], Siri Tähtinen, Cristian Capasso, Sara Feola, Beatriz Martins, Jacopo Chiaro, Firas Hamdan, Otto K. Kari, Joseph Ndika, Erko Ylösmäki, Karita Peltonen, Leena Ylösmäki, Harri Alenius, Arto Urtti, Jouni T. Hirvonen, Hélder A. Santos, Vincenzo Cerullo**

Reviewer #2:

The authors have done a commendable job of addressing the comments from the previous round of review. They have provided thoughtful responses and clarifications, which are much appreciated.

Re: We are deeply grateful to the reviewer for the time spent reading the revised version of the manuscript and for the insightful comments in all the stages of revision.

- 1. An issue remains with the response to the query regarding the neutralizing effect of serum obtained from mice exposed to the virus (Reviewer 2, Question 3e). In theory, the membrane coating around the virus should preclude drops in infectivity caused by neutralizing antibodies, whereas the data clearly shows that these antibodies are able to neutralize the coated ExtraCRAd formulation. As noted by the Reviewer, it cannot be claimed that 1:10 is very concentrated, particularly given that whole serum (as would be encountered in a patient) is 10 times more concentrated. While this may not affect the authors' in vivo studies in the present work, this does raise concerns about the real-world applicability of the platform. Does this mean that the treatment would only be effective in patients naïve to the virus? More discussion should be included along these lines.**

Re: We thank the reviewer for the important comment. We agree on the issue raised by the reviewer and we discussed about it at page 16. However, in our experimental setup, the mice were preimmunized 4 times in a month (once every week), resulting in an artificial immunological pattern which may not resemble the patient status. This served mainly as a proof of concept for our technology and it will be further investigated in future.